# More Than a Snapshot: Forcing Temporal Reasoning in Video Segmentation

## Abstract

Video Reasoning Segmentation (VRS) inherits the settings of reasoning based on world knowledge and spatial contents, lacking queries demanding temporal reasoning according to the unique temporal dynamics of videos. To bridge the gap, we introduce TempVRS, a large-scale Temporal Video Reasoning Segmentation dataset containing 30k videos and 200k queries injecting temporal dynamics. Moreover, existing VRS methods commonly employ a three-stage paradigm: keyframe selection, reasoning and propagation. However, such paradigm not only neglects temporal dynamics inherent in videos which results in non-negligible deviations of keyframe selections, but also hinders video understanding, leading to the degradation of video reasoning into isolated keyframe analysis. To address the defects of such paradigm, we propose a temporal video reasoning segmentation method to stimulate the inherent temporal-reasoning capabilities of multimodal large language model. Through interleaving uniform-sampled video frames across spatial dimension and explicitly injecting spatiotemporal distribution, our 4B-method can achieve comparable performance with Sa2VA-8B under the same inference settings, significantly improving the accuracy when evaluated on existing referring/reasoning video segmentation benchmarks (e.g., $5.5\%$ and $3.4\%$ increases compared to Sa2VA-4B on MeViS and ReVOS).

## 1 Introduction

Recently, the task of Image Reasoning Segmentation has emerged to assist open-world segmentation by leveraging Large Language Models (LLMs). Through incorporating an special token <SEG> to bridge LLMs (Liu et al., 2024a) and segmentation models (e.g. SAM (Kirillov et al., 2023)), LISA (Lai et al., 2024) can segment target objects after reasoning the queries with world knowledge. During the rapid expansion from image to video, the task of Video Reasoning Segmentation (VRS) retains the basic setting of reasoning with world knowledge such as the reasoning-type queries in ReVOS dataset (Yan et al., 2024), neglecting to incorporate the unique temporal dynamics of videos in the query setting. The lack of temporal dynamics leads to representative methods adopt a short-cut paradigm including selecting keyframes, reasoning based on keyframes and query, propagating keyframe segmentation to the whole video.

To dig into the effect of temporal dynamics for existing keyframe-based VRS methods (Yan et al., 2024; Gong et al., 2025; Lin et al., 2025; Yuan et al., 2025), we perform exploratory experiments by manually incorporating temporal dynamics on queries and retesting the accuracy. As shown in Fig. 1 (a), we select 100 videos from val-set of ReVOS (Yan et al., 2024) and add another object descriptions to clarify the temporal location of the target objects through the temporal conjunction such as before, after, when, etc. After clarifying the impacts on accuracy before and after queries change without changing selected keyframes as illustrated in Fig. 1 (b), we evaluate existing methods (Gong et al., 2025; Lin et al., 2025; Yuan et al., 2025) re-selecting keyframes according to queries with temporal dynamics. The non-negligible declines in accuracy indicate the significant keyframe selection deviations under the premise that the target objects of queries remain unchanged.

When facing queries with temporal dynamcis, the deviations of selected keyframes result in semantic deviations to objects of frames used for temporal location when existing VRS methods generate <SEG> tokens. The results of experiments clearly reveal the **Temporal-Dynamic Dilemma**: Frames matching with queries are required and selected for temporal location whereas existing meth-

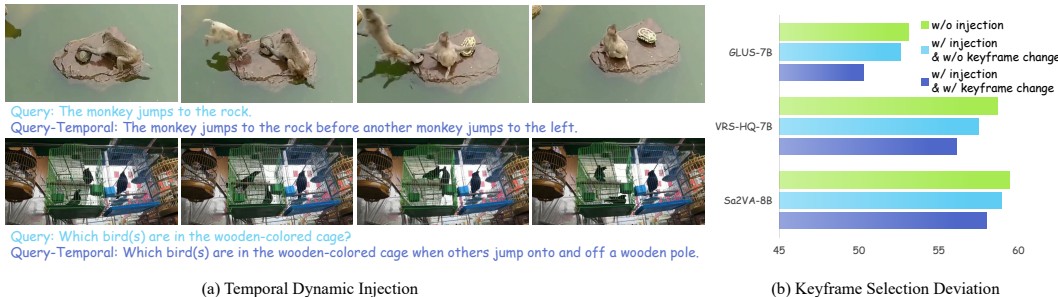

(a) Temporal Dynamic Injection

(b) Keyframe Selection Deviation

Figure 1: Deviations of keyframe selection when facing queries with temporal dynamics.

ods lacking temporal reasoning cannot understand and confirm frames at which timestamps are key for segmenting target objects. This temporal-dynamic dilemma motivates us to explore temporal-spatial reasoning capabilities inherent in MLLMs to determine which frames and which objects in these frames are corresponding to queries with temporal dynamcis., rather than simply matching between visual and linguistic representations.

To make up for the shortage of temporal dynamics, we construct a large-scale Temporal Video Reasoning Segmentation dataset, termed as TempVRS, which contains 31701 videos selected from SAV and about 200,000 queries reflecting temporal dynamics. Concretely, we desgin three types of query templates and equip each query with ground-truth temporal location of target objects in videos. As shown in Fig. 2, three templates are defined as follows: (1) **Temporal Order**: Connect different descriptions of objects with temporal conjunctinos such as before, after, etc. (2) **Temporal Event**: Merge multiple event descriptions of the same target object at different timestamps into one query. (3) **Temporal Count**: Query the target objects based on counting in temporal dimension. Besides, to further explore the performance of VRS methods in temporal reasoning on a wider temporal dimension, we additionally construct a long-term temporal reasoning subset containing 200 videos where each video contains thousands of frames.

To address temporal dynamics, we design an Temporal Video Reasoning Segmentation method that fully exploits the intrinsic reasoning capabilities of MLLMs. Without keyframe selections, we arrange the uniform-sample slow-fast frames into a spatial-interleaved pattern to maintain temporal order and input them into LLMs to ensure the spatial richness. Through explicitly performing spatiotemporal supervision during training, our method learns the intrinsic correlation between frames and query based on spatiotemporal reasoning. Extensive experimental results on existing referring/reasoning video object segmentation datasets illustrate that our proposed method can significantly improve the performance. Moreover, the benchmark results on our proposed TempVRS dataset indicate that blindly performing keyframe selections without temporal reasoning can seriously mislead the segmentation of target objects. Furthermore, the benchmark discussions on long-term videos can stimulate explorations on how to deal with thousands frames in long-term videos. Our contributions can be summarized as follows:

- We reveal that temporal dynamics injected into queries can result in significant deviations of keyframe selection, which makes higher demands for video reasoning segmentation.

- We construct a large-scale Temporal Video Reasoning Segmentation dataset containing 30k videos and 200k queries where temporal reasoning based on video understanding becomes the key to confirm target objects.

- We design a temporal-reasoning video segmentation method to invoke the spatiotemporal reasoning ability of MLLMs through learning spatiotemporal distributions. With extensible spatial-interleaved architecture, our method can handle long-term video segmentation.

- Extensive experiments demonstrate temporal-reasoning can effectively promote video understanding and improve segmentation performance. Further discussions on how to deal with long-term video reasoning segmentation provides exploration room in VRS field.

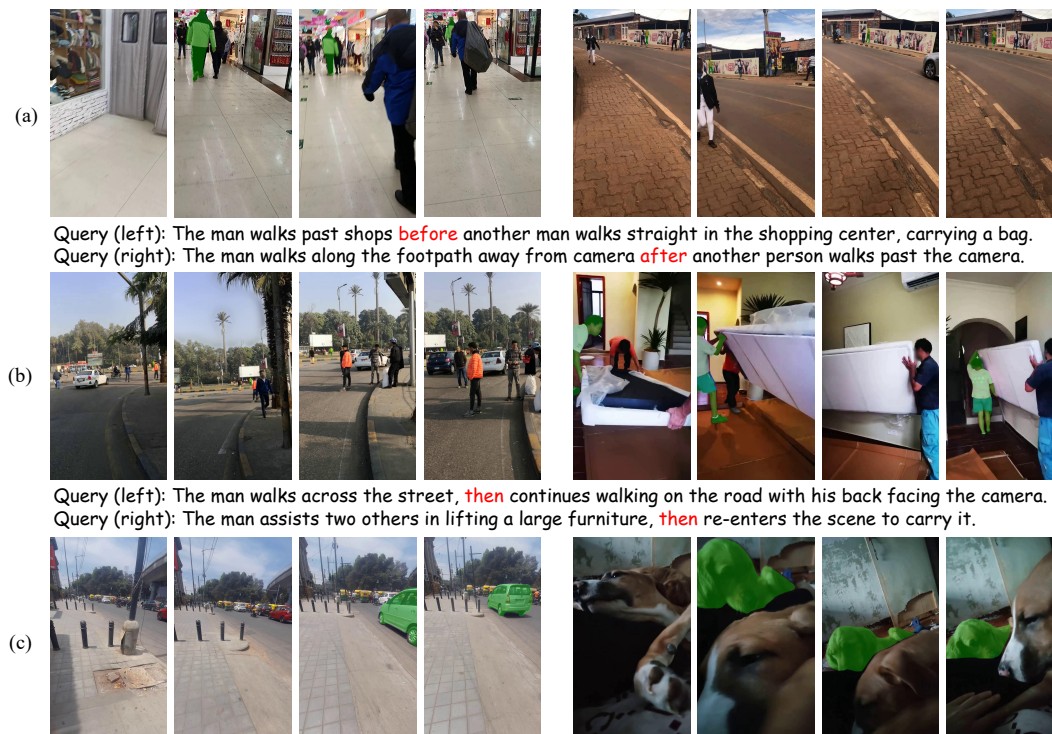

(a)

Query (left): The man walks past shops before another man walks straight in the shopping center, carrying a bag.
Query (right): The man walks along the footpath away from camera after another person walks past the camera.

(b)

Query (left): The man walks across the street, then continues walking on the road with his back facing the camera.
Query (right): The man assists two others in lifting a large furniture, then re-enters the scene to carry it.

(c)

Query (left): The last vehicle moving on the road along with other vehicles away from the camera.
Query (right): The second dog that appears in the camera is lying on the bed.

Figure 2: Examples of three temporal templates including (a) temporal order, (b) temporal event and (c) temporal count. Target objects in frames are marked in green.

## 2 RELATED WORKS

### 2.1 REFERRING VIDEO OBJECT SEGMENTATION

Referring Video Object Segmentation aims at segmenting target objects based on given expressions. The seminal transformer-based methods MTTR (Botach et al., 2022) and ReferFormer (Wu et al., 2022), regarding the RVOS task as a sequence prediction problem and determining the sequences of target objects given a text query, takes the segmentation performance to a new level. Based on this effective transformer architecture, RVOS methods including HTML (Han et al., 2023), TempCD (Tang et al., 2023b), SOC (Luo et al., 2024), SgMg (Miao et al., 2023), SSA (Pan et al., 2025), ReferDINO (Liang et al., 2025b) enhances visual-language alignment from different perspectives. Moreover, LMPM (Ding et al., 2023) raises concerns about motion of objects, DsHmp (He & Ding, 2024) and DMVS (Fang et al., 2025) decouple the static and motion informations to further enhance motion comprehension. Although these RVOS methods make many attempts in cross-modal interaction and fusion, the overall segmentation performance is still limited by the understanding ability of the text encoder (e.g., RoBERTa (Liu, 2019)).

### 2.2 REASONING SEGMENTATION

Recently, LISA (Lai et al., 2024) introduces image reasoning segmentation task which segments target objects based on the language understanding ability of LLMs, thereby the descriptions to objects in RVOS task can be extended to complex queries which need to reason with world knowledge. GSVA (Xia et al., 2024) and Seg-LLM (Wang et al., 2024) further expand the settings of image reasoning segmentation by introducing object rejection and multi-round interaction. Inheriting the basic settings of image reasoning segmentation, VISA (Yan et al., 2024) further introduces video reasoning segmentation task where the expressions in RVOS dataset are replaced with more general queries. Based on VISA, MLLMs including GLUS (Lin et al., 2025), VRS-HQ (Gong et al., 2025) are proposed to improve the ability of keyframe selection with global-local sampling and mask prop-

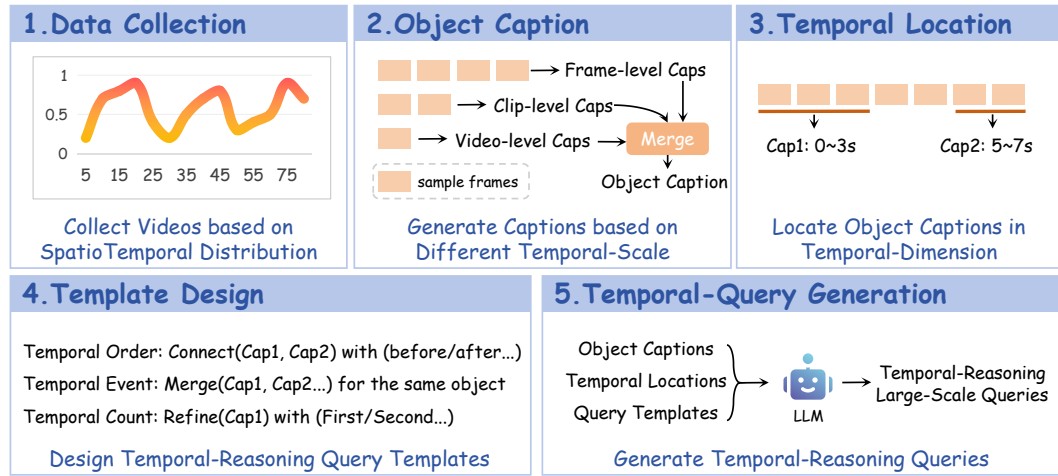

Figure 3: Pipeline of our dataset construction.

agation. Sa2VA (Yuan et al., 2025) is constructed to support both segmentation and chat tasks, thus its ability for expression understanding has not deteriorated too much and achieves the highest performance in VRS task. However, these VRS methods which lack the ability of temporal reasoning, produce serious keyframe selection deviations when faced with queries full of temporal dynamics, limiting the effects of subsequent reasoning. To bridge the gap of temporal reasoning, we construct a temporal video reasoning segmentation dataset and a temporal reasoning method beyond keyframe selection to address temporal dynamics.

## 3 TEMPVRS

### 3.1 TEMPVRS DATASET AND BENCHMARK

**Dataset Construction.** The construction process of our proposed temporal video reasoning segmentation dataset (TempVRS) consists of five steps as shown in Fig. 3: **(1) Data Colletction.** From the large-scale video object segmentation dataset SAV (Ravi et al., 2024), we collect videos by calculating the spatiotemporal distribution of target objects based on their high-quality mask annotations. Specifically, for one target object, we count the percentage of target appearances in videos. We filter out videos where all target objects have appearance percentage over $95\%$ in entire video and finally confirm 31701 videos to construct our TempVRS dataset. **(2) Object Caption.** We generate captions of each target object by sampling frames in different temporal-scale (from frame, clip to video). During captioning, we also use mask of target objects to generate object-level and scene-level captions following Sa2VA (Yuan et al., 2025). **(3) Temporal Location.** For each caption generated in step 2, we further equip these captions with temporal locations. Specifically, as we can clearly identify in which frames of videos the target objects appear based on the mask, assigning the start and end timestamps becomes a matching between appearance clips and captions. We adopt LLM (Hurst et al., 2024) to estimate the correlation between clip-level captions generated in step 2 and final captions to confirm the temporal locations. **(4) Template Design.** We design three templates including temporal order, temporal event and temporal count to enrich the temporal-reasoning queries. For different target objects in the same video, we connect their captions with temporal conjunctions such as before, after, etc. For the same target object, we merge captions with different temporal locations to increase the temporal dynamics of queries. Through counting the number of appearance of different target objects, we refine captions with first, second, last, etc. **(5) Temporal-Query Generation.** After designing templates of prompts, we instantiate templates with object captions and temporal locations, then ask LLM (Hurst et al., 2024) to generate about 200k temporal-reasoning queries. The statistics of our proposed TempVRS dataset are listed in Appendix A.2.

**Evaluation Benchmark.** We split our large-scale TempVRS datasets into training and test subset containing 30701 and 1000 videos respectively. The average frame number of videos in our TempVRS dataset is 83, which is relatively few for our proposed method with slow-fast architecture that usually processes long-term videos. Therefore, we additionally collect 200 videos with 1248 aver-

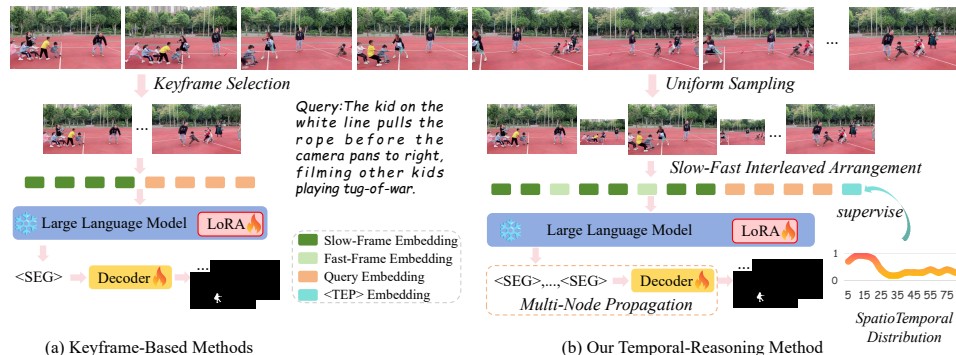

Figure 4: Architecture comparison between (a) existing Keyframe-Based Methods (Yan et al., 2024; Lin et al., 2025; Yuan et al., 2025) and (b) Our proposed Temporal-Reasoning Method.

age frame (we unify the sampling rate to 6 fps) from the existing video object segmentation/tracking datasets (Tang et al., 2023a; Hong et al., 2023; 2024; Hu et al., 2023), then generate temporal-queries with our dataset construction pipeline, finally we obtain 1000 short-term videos and 200 long-term videos. The queries for target objects in these 1200 videos are carefully filtered by human annotators and retained 7312, three types of temporal queries are balanced to 32%/39%/29% (Order/Event/Count). The subset of long-term videos are used for zero-shot evaluation. We select 150 short-term videos and 50 long-term videos to construct the validation set, and the final evaluation benchmark set consists of remaining 850 short-term and 150 long-term videos.

## 3.2 BEYOND KEYFRAME SELECTION

Through embedding hidden states into a special token <SEG> in the output, existing keyframe-based reasoning video object segmentation methods as shown in Fig. 4 (a) can generate semantic prompts for segmentation decoders. As the reasoning process corresponding to <SEG> is implicitly regulated by supervised training on the generated mask, the spatiotemporal reasoning objective required for reasoning video object segmentation cannot be directly optimized. To this end, we append a special token <TEP> at the end of input to MLLM, then explicitly supervise the spatiotemporal reasoning by extracting the spatiotemporal distribution of target objects in videos during training.

**Condensing Temporal Dynamics.** As shown in Fig. 4 (b), given a video and query, we firstly uniform-sample $K$ frames of videos, then arrange the encoded embeddings of $K$ frames according to their temporal order, finally append embeddings of query and special token <TEP> as the input of LLM. Different from training a keyframe selector (Hu et al., 2025; Buch et al., 2025) with similar learnable embeddings to boost video understanding based on the selected keyframes in two stages, our method directly generates <SEG> based on the condensed summary of the temporal dynamics in relation to the query encapsulated in the embedding of special token <TEP>, thereby our method can leverage the condensed temporal informations to produce temporal-aware reasoning outputs.

**Defects of Uniform Sampling.** As the number of video frames increases, the number of uniform-sampling frames cannot increase synchronously due to memory limitation, leading to no longer covering the keyframes that are relevant to the input query which are useful for following reasoning. The field of long-video understanding usually adopts the strategy of constructing Slow-Fast Architecture (Xu et al., 2024; 2025) to increase the visibility of video frame contents during LLM reasoning at a lower cost compared to directly increasing the number of high-resolution frames. In these Slow-Fast architecture, visual embeddings in slow path and fast path are aggregated into two separate areas of the input tokens to LLM. This separate arrangement strategy disrupt the original temporal order between different frames, which makes against temporal reasoning.

**Slow-Fast Interleaved Architecture.** To this end, we interleave the visual embeddings in the input across spatiotemporal dimensions as shown in Fig. 4. Specifically, we firstly uniform-sample $K_s$ slow-frames with high resolutions from videos, then we uniform-sample $K_f$ fast-frames between every two adjacent slow-frames, finally we arrange the embeddings of slow-fast frames with high-low resolutions in the input according to their original temporal order in the video. There-

fore, spatiotemporal reasoning supervised through <TEP> can be correctly achieved based on the embeddings of slow-fast frames corresponding to the temporal order of query.

**Necessity of Fast-frames.** During inference, based on the spatiotemporal correlation vector decoded from the embedding of special token <TEP>, we select top-k frames as multiple nodes to propagate their decoded mask to the whole videos. The strategies of generating one <SEG> then copying it for multiple-nodes or directly generating multiple different <SEG> when facing videos of different lengths are discussed in Sec. 4. In fact, reasoning based on fast-frames with lower details of target objects inherently reduce the accuracy. For short-term videos in existing referring video object datasets, there is no need to introduce fast-frames as these short videos are sufficient to be covered by slow-frames of uniform sampling. **However, when facing long-term videos, the positive benefits of using fast-frames to cover the frames where the target appears far outweigh the negative benefits caused by the loss of details.**

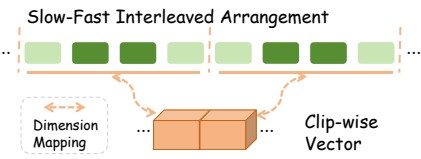

Figure 5: Illustrations of dimension mapping during supervision.

**Clip-wise Supervision.** Besides commonly used text regression loss $L_{text}$ and segmentation loss $L_{mask}$ including pixel-wised cross-entropy loss and dice loss, we adopt MLP to decode the embedding of special token <TEP> and supervise it with cross-entropy loss $L_{ce}$. Compared to decoding into a vector with the same dimension as the number of input frames, we further compress the dimension based on the arrangement of slow-fast frames as shown in Fig. 5. If we predict the spatiotemporal correlation between fast-frames and the input query in a frame-by-frame mode, uncontrollable noise will be introduced due to the relatively small number of embedding tokens of fast-frames, which is not conducive to the subsequent selection of video frames for mask propagation.

## 4 EXPERIMENTS

### 4.1 EXPERIMENT DETAILS

Following Sa2VA (Yuan et al., 2025), we adopt InternVL2-5 (Chen et al., 2024) and SAM2 (Ravi et al., 2024) in our TempVRS method. **It should be noted that our training resources only support training 4B MLLMs, however, our TempVRS-4B method can outperform 7B and even 13B MLLMs in existing video segmentation benchmarks. We will open-source model, dataset for training models with larger parameter-scales.**

Following existing VRS methods (Yan et al., 2024; Lin et al., 2025; Gong et al., 2025; Yuan et al., 2025), we train our TempVRS with image QA (Liu et al., 2024b), video QA (Jin et al., 2024), image segmentation (Kazemzadeh et al., 2014; Yu et al., 2016) and video segmentation datasets (Ding et al., 2023; Seo et al., 2020; Yan et al., 2024). When training with video segmentation datasets, we fisrt estimate spatiotemporal distribution of these videos with the strategy in our dataset construction pipeline, then train our method with these auto-generated distributions.

During evaluation, following (He & Ding, 2024; Liang et al., 2025b; Yan et al., 2024; Yuan et al., 2025), we adopt commonly used metrics including region similarity ($\mathcal{J}$), contour accuracy ($\mathcal{F}$) and their mean ($\mathcal{J}\&\mathcal{F}$). More training and inference details are listed in the Appendix A.3.

### 4.2 MAIN RESULTS

Compared to existing VRS methods (Yan et al., 2024; Zhu et al., 2023; Lin et al., 2025; Gong et al., 2025; Yuan et al., 2025), our TempVRS method achieves the best performance when evaluating on ReVOS (Yan et al., 2024), Ref-YouTube-VOS (Seo et al., 2020), Ref-DAVIS17 (Khoreva et al., 2019) and MeViS (Ding et al., 2023) datasets as shown in Tab. 1 and Tab. 2 . For example, compared to Sa2VA-4B which adopts the same MLLM (Chen et al., 2024) as our method, our TempVRS achieves $5.5\%$ and $3.4\%$ improvements in terms of $\mathcal{J}\&\mathcal{F}$ on MeViS test set and ReVOS validation set. When evaluating on Ref-DAVIS17, MeViS and ReVOS datasets, the performance of our 4B method can surpass the one of Sa2VA-8B, which effectively illustrates the key of temporal reasoning in video reasoning segmentation process. We further train our method with our proposed TempVRS dataset, and the results clearly show the significant accuracy improvements.

Table 1: Comparison of our method with SOTA methods when evaluating on ReVOS (Yan et al., 2024) dataset. ∗ denotes additionally trained with our TempVRS dataset.

| Method | Backbone | referring | | | reasoning | | | overall | | | $\mathcal{R}$ |
|---|---|---|---|---|---|---|---|---|---|---|---|
| | | $\mathcal{J}\&\mathcal{F}$ | $\mathcal{J}$ | $\mathcal{F}$ | $\mathcal{J}\&\mathcal{F}$ | $\mathcal{J}$ | $\mathcal{F}$ | $\mathcal{J}\&\mathcal{F}$ | $\mathcal{J}$ | $\mathcal{F}$ | |
| Methods without LLMs | | | | | | | | | | | |
| MTTR (Botach et al., 2022) | Video-Swin-T | 30.0 | 29.8 | 30.2 | 21.0 | 20.4 | 21.5 | 25.5 | 25.1 | 25.9 | 5.6 |
| LMPM (Ding et al., 2023) | Video-Swin-T | 34.1 | 29.0 | 39.1 | 18.8 | 13.3 | 24.3 | 26.4 | 21.2 | 31.7 | 3.2 |
| ReferFormer (Wu et al., 2022) | Video-Swin-B | 32.7 | 31.2 | 34.3 | 23.4 | 21.3 | 25.6 | 28.1 | 26.2 | 29.9 | 8.8 |
| Methods with 7B-13B LLMs | | | | | | | | | | | |
| LISA (Lai et al., 2024) | LLaVA-7B | 45.7 | 44.3 | 47.1 | 36.1 | 33.8 | 38.4 | 40.9 | 39.1 | 42.7 | 9.3 |
| LISA (Lai et al., 2024) | LLaVA-13B | 46.6 | 45.2 | 47.9 | 36.7 | 34.3 | 39.1 | 41.6 | 39.8 | 43.5 | 8.6 |
| TrackGPT (Zhu et al., 2023) | LLaVA-7B | 48.2 | 46.7 | 49.7 | 39.0 | 36.8 | 41.2 | 43.6 | 41.8 | 45.5 | 11.6 |
| TrackGPT (Zhu et al., 2023) | LLaVA-7B | 49.5 | 48.3 | 50.6 | 40.5 | 38.1 | 42.9 | 45.0 | 43.2 | 46.8 | 12.8 |
| VISA (Yan et al., 2024) | Chat-UniVi-7B | 50.9 | 49.2 | 52.6 | 43.0 | 40.6 | 45.4 | 46.9 | 44.9 | 49.0 | 15.5 |
| VISA (Yan et al., 2024) | Chat-UniVi-13B | 57.4 | 55.6 | 59.1 | 44.3 | 42.0 | 46.7 | 50.9 | 48.8 | 52.9 | 14.5 |
| GLUS (Lin et al., 2025) | LLaVA-7B | 58.3 | 56.0 | 60.7 | 51.4 | 48.8 | 53.9 | 54.9 | 52.4 | 57.3 | 17.9 |
| VRS-HQ (Gong et al., 2025) | Chat-UniVi-7B | 62.1 | 59.8 | 64.5 | 56.1 | 53.5 | 58.7 | 59.1 | 56.6 | 61.6 | 19.7 |
| VRS-HQ (Gong et al., 2025) | Chat-UniVi-13B | 63.3 | 61.1 | 65.5 | 56.8 | 54.1 | 59.4 | 60.0 | 57.6 | 62.5 | 18.9 |
| Sa2VA (Yuan et al., 2025) | InternVL2.5-8B | 66.0 | 63.3 | 68.7 | 60.3 | 57.2 | 63.3 | 63.1 | 60.2 | 66.0 | 16.9 |
| Methods with 4B LLMs | | | | | | | | | | | |
| Sa2VA (Yuan et al., 2025) | InternVL2.5-4B | 63.3 | 60.9 | 65.7 | 56.7 | 53.9 | 59.5 | 60.0 | 57.4 | 62.6 | 21.5 |
| TempVRS (Ours) | InternVL2.5-4B | 66.2 | 63.5 | 68.9 | 60.7 | 57.6 | 63.8 | 63.4 | 60.6 | 66.3 | 21.6 |
| TempVRS* (Ours) | InternVL2.5-4B | 68.4 | 65.9 | 70.9 | 62.0 | 59.0 | 65.0 | 65.2 | 62.4 | 67.9 | 20.4 |

Table 2: Comparison of our method with SOTA methods when evaluating on MeViS (Ding et al., 2023), Ref-YouTube-VOS (Seo et al., 2020), and Ref-DAVIS17 (Khoreva et al., 2019) datasets. ∗ denotes additionally trained with our TempVRS dataset.

| Method | Backbone | MeViS | | | Ref-YouTube-VOS | | | Ref-DAVIS17 | | |
|---|---|---|---|---|---|---|---|---|---|---|
| | | $\mathcal{J}\&\mathcal{F}$ | $\mathcal{J}$ | $\mathcal{F}$ | $\mathcal{J}\&\mathcal{F}$ | $\mathcal{J}$ | $\mathcal{F}$ | $\mathcal{J}\&\mathcal{F}$ | $\mathcal{J}$ | $\mathcal{F}$ |
| Methods without LLMs | | | | | | | | | | |
| MTTR (Botach et al., 2022) | Video-Swin-T | 30.0 | 28.8 | 31.2 | 55.3 | 54.0 | 56.6 | - | - | - |
| ReferFormer (Wu et al., 2022) | Video-Swin-B | 31.0 | 29.8 | 32.2 | 62.9 | 61.3 | 64.6 | 61.1 | 58.1 | 64.1 |
| OnlineRefer (Wu et al., 2023) | Swin-L | - | - | - | 63.5 | 61.6 | 65.5 | 64.8 | 61.6 | 67.7 |
| LoSh (Yuan et al., 2024) | Video-Swin-B | - | - | - | 67.2 | 65.4 | 69.0 | 64.3 | 61.8 | 66.8 |
| DsHMP (He & Ding, 2024) | Video-Swin-B | 46.4 | 43.0 | 49.8 | 67.1 | 65.0 | 69.1 | 64.9 | 61.7 | 68.1 |
| ReferDINO (Liang et al., 2025b) | Video-Swin-B | 49.3 | 44.7 | 53.9 | 69.3 | 67.0 | 71.5 | 68.9 | 65.1 | 72.9 |
| Methods with 7B-13B LLMs | | | | | | | | | | |
| LISA (Lai et al., 2024) | LLaVA-7B | 37.2 | 35.1 | 39.4 | 53.9 | 53.4 | 54.3 | 64.8 | 62.2 | 67.3 |
| LISA (Lai et al., 2024) | LLaVA-13B | 37.9 | 35.8 | 40.0 | 54.4 | 54.0 | 54.8 | 66.0 | 63.2 | 68.8 |
| TrackGPT (Zhu et al., 2023) | LLaVA-7B | 40.1 | 37.6 | 42.6 | 56.4 | 55.3 | 57.4 | 63.2 | 59.4 | 67.0 |
| TrackGPT (Zhu et al., 2023) | LLaVA-7B | 41.2 | 39.2 | 43.1 | 59.5 | 58.1 | 60.8 | 66.5 | 62.7 | 70.4 |
| VISA (Yan et al., 2024) | Chat-UniVi-7B | 43.5 | 40.7 | 46.3 | 61.5 | 59.8 | 63.2 | 69.4 | 66.3 | 72.5 |
| VISA (Yan et al., 2024) | Chat-UniVi-13B | 44.5 | 41.8 | 47.1 | 63.0 | 61.4 | 64.7 | 70.4 | 67.0 | 73.8 |
| GLUS (Lin et al., 2025) | LLaVA-7B | 51.3 | 48.5 | 54.2 | 67.3 | 65.5 | 69.0 | - | - | - |
| VRS-HQ (Gong et al., 2025) | Chat-UniVi-7B | 50.6 | 47.6 | 53.7 | 70.4 | 68.3 | 72.5 | 76.0 | 72.6 | 79.4 |
| VRS-HQ (Gong et al., 2025) | Chat-UniVi-13B | 50.9 | 48.0 | 53.7 | 71.0 | 69.0 | 73.1 | 74.4 | 71.0 | 77.9 |
| Sa2VA (Yuan et al., 2025) | InternVL2.5-8B | 53.6 | 50.4 | 56.8 | 72.4 | 70.0 | 74.8 | 76.5 | 72.3 | 80.8 |
| Methods with 4B LLMs | | | | | | | | | | |
| VideoLISA (Bai et al., 2024) | LLaVA-Phi-3-V | 44.4 | 41.3 | 47.6 | 63.7 | 61.7 | 65.7 | 68.8 | 64.9 | 72.7 |
| Sa2VA (Yuan et al., 2025) | InternVL2.5-4B | 48.3 | 45.2 | 51.3 | 71.7 | 69.5 | 74.0 | 73.9 | 69.7 | 78.1 |
| TempVRS (Ours) | InternVL2.5-4B | 53.8 | 50.6 | 57.1 | 71.9 | 69.7 | 74.1 | 76.8 | 72.7 | 80.9 |
| TempVRS* (Ours) | InternVL2.5-4B | 54.2 | 50.8 | 57.6 | 72.1 | 69.8 | 74.4 | 79.0 | 74.8 | 83.2 |

We evaluate open-sourced VRS methods (Yan et al., 2024; Lin et al., 2025; Gong et al., 2025; Yuan et al., 2025) on our proposed large-scale TempVRS benchmark. As shown in Tab. 3, without finetuning with our TempVRS dataset, our method achieves the best performance when facing both short-term and long-term videos. Moreover, our method

Table 3: Benchmark on our proposed TempVRS dataset. ∗ denotes additionally trained with our TempVRS dataset..

| Method | Test(Short) | | | Test(Long, zero-shot) | | |
|---|---|---|---|---|---|---|
| | $\mathcal{J}\&\mathcal{F}$ | $\mathcal{J}$ | $\mathcal{F}$ | $\mathcal{J}\&\mathcal{F}$ | $\mathcal{J}$ | $\mathcal{F}$ |
| VISA-7B (Yan et al., 2024) | 33.1 | 31.0 | 35.2 | 21.1 | 17.4 | 24.8 |
| GLUS-7B (Lin et al., 2025) | 37.8 | 35.7 | 39.9 | 24.5 | 20.9 | 28.1 |
| VRS-HQ-7B (Gong et al., 2025) | 38.3 | 36.2 | 40.4 | 23.6 | 20.0 | 27.2 |
| Sa2VA-8B (Yuan et al., 2025) | 40.6 | 38.9 | 42.3 | 25.8 | 22.1 | 29.5 |
| TempVRS (Ours) | 42.7 | 40.3 | 45.1 | 30.9 | 27.7 | 34.1 |
| TempVRS* (Ours) | 48.9 | 47.1 | 50.7 | 31.3 | 28.2 | 34.4 |

achieves more accuracy improvements on the evaluation of long-term videos than on short-term videos. When faced with long-term videos with hundreds of frames, existing keyframe-based methods encounter more significant deviations during keyframe selection, resulting in a striking decline in the segmentation performance. After finetuning on our TempVRS train set which consists of short-term videos, our method can achieve remarkable improvements when evaluated on the subset of short-term videos. Meanwhile, the accuracy of zero-shot evaluation on long-term videos after finetuning on our proposed TempVRS dataset has also been improved which illustrates the robustness of our method.

## 4.3 ABLATION STUDIES

**When do we need fast-frames?** We conduct ablation studies on our slow-fast interleaved architecture. When facing short-term videos, we increase the number of slow-frames (from 10 to 15) without fast-frames and the corresponding accuracy improvements on short-term evaluation are negligible. Therefore, we can conclude that slow-frames have a high coverage of short-term videos which are sufficient to handle short-term video reasoning. Moreover,

Table 4: Ablation studies on number of slow/fast frames ($K_s/K_l$) and number of fast-frame embedding tokens ($N_f$).

| Setting | Val (Short) | | | Val (Long, zero-shot) | | |
|---|---|---|---|---|---|---|
| | $\mathcal{J}\&\mathcal{F}$ | $\mathcal{J}$ | $\mathcal{F}$ | $\mathcal{J}\&\mathcal{F}$ | $\mathcal{J}$ | $\mathcal{F}$ |
| $K_s = 10, K_f = 0$ | 46.4 | 45.5 | 47.3 | 34.6 | 31.7 | 37.5 |
| $K_s = 15, K_f = 0$ | 46.6 | 45.7 | 47.5 | 35.2 | 32.1 | 38.3 |
| $K_s = 10, K_f = 10, N_f = 16$ | 41.3 | 40.8 | 41.8 | 34.9 | 32.0 | 37.8 |
| $K_s = 10, K_f = 10, N_f = 64$ | 43.5 | 42.4 | 44.6 | 36.7 | 34.3 | 39.1 |
| $K_s = 10, K_f = 20, N_f = 64$ | 43.9 | 42.7 | 45.1 | 38.0 | 36.1 | 39.9 |

we set the number of slow-frames and fast-frames both to 10. The results show that the introduction of fast-frames makes the reasoning process based on visual tokens with more loss of details, leading to accuracy decline. As the number of fast-frame tokens decreases (from 64 to 16), the accuracy decreases further. When facing long-term videos, the slow-fast setting greatly enhances the coverage of target appearance frames without introducing excessive computational burden (compared to only using more slow-frames), which achieves higher accuracy compared to only-slow setting.

**What is the effect of clip-wise supervision?** Compared to supervision in frame-by-frame mode, our clip-by-clip supervision mode reduces the noise during association process between fast-frames and query, increasing the accuracy as shown in Tab. 5, especially achieving more significant improvements when evaluated on long-term videos (from 33.5 to 38.0 in terms of $\mathcal{J}\&\mathcal{F}$).

**How effective is it to replace uniform sampling with keyframe selection?** We further replace the uniform sampling with selecting keyframes following (Gong et al., 2025). As shown in Tab. 5, we replace the uniform-sampling slow-frames with selected keyframes, the performances on short-term videos increase while on long-term videos decrease. The results on short-term

Table 5: Ablation studies on <TEP> supervision (frame-wise or clip-wise) and sampling strategies of slow-frames (uniform sampling or keyframe selection).

| Setting | Val (Short) | | | Val (Long, zero-shot) | | |
|---|---|---|---|---|---|---|
| | $\mathcal{J}\&\mathcal{F}$ | $\mathcal{J}$ | $\mathcal{F}$ | $\mathcal{J}\&\mathcal{F}$ | $\mathcal{J}$ | $\mathcal{F}$ |
| frame/Uniform | 41.3 | 39.6 | 43.0 | 33.5 | 30.9 | 36.1 |
| clip/Uniform | 43.9 | 42.7 | 45.1 | 38.0 | 36.1 | 39.9 |
| clip/Keyframe | 44.1 | 42.8 | 45.5 | 34.6 | 32.7 | 36.5 |

videos indicate that our method can get rid of temporal-dynamic dilemma: as input frames match queries better, our method can further improve performance with spatial-temporal reasoning capabilities. When facing long-term videos, the selection of keyframes is more susceptible to interference with frames containing similar objects, thereby the video frames that could have covered the appearance of target objects through the setting of slow-fast frames are replaced by the wrong selected keyframes, leading to significant decline (from 38.0 to 34.6 in terms of $\mathcal{J}\&\mathcal{F}$).

## 4.4 FAILURE CASES

We visualize the failure cases during evaluation as shown in Fig. 6. The two cases represent two extremely difficult challenges in our proposed TempVRS dataset including tiny object and long query. For example, the target object in case (a) is the tiny person in the distant background which needs to zoom in to confirm the target. In case (b), due to we concatenate events from different

Figure 6: Failure cases representing (a) tiny object and (b) long query. In each case, the top and bottom rows show the frames marked with ground-truth mask and predicted mask respectively.

temporal periods of the target object into one query, it contains too many words which is challenging for VRS methods to understand the query and predict the correct target object.

## 5 DISCUSSIONS

**Distribution imbalance.** For existing video reasoning segmentation datasets (Seo et al., 2020; Ding et al., 2023; Yan et al., 2024), the proportion of frames in which target objects appear to total frames is too high. Directly using these imbalanced data for training leads to unsatisfactory <TEP> learning results. Through balancing positive-negtive samples during training (details of balance are listed in Appendix A.3), the accuracy is improved as shown in Tab. 6. The balance of spatiotemporal distribution to boost reasoning deserves further research.

**Multiple <SEG> tokens.** We conduct experiments on <SEG> predictions in two modes: 1) generate one <SEG> and repeat it for multiple frames, 2) generate multiple <SEG>. During evaluation on long-term videos, we find the appearance of the same target objects corresponding to the query in the video are inconsistent. Therefore, we generate multiple <SEG> tokens (same

Table 6: Discussions on distribution imbalance (Direct/Balance Training) and multiple <SEG> (Repeat one or generate Different)

| Setting | Val (Short) | | | Val (Long, zero-shot) | | |
|---|---|---|---|---|---|---|
| | $\mathcal{J}\&\mathcal{F}$ | $\mathcal{J}$ | $\mathcal{F}$ | $\mathcal{J}\&\mathcal{F}$ | $\mathcal{J}$ | $\mathcal{F}$ |
| Direct/Different | 38.7 | 36.2 | 41.2 | 31.6 | 28.5 | 34.7 |
| Balance/Repeat | 44.7 | 43.1 | 46.3 | 36.2 | 35.8 | 36.6 |
| Balance/Different | 43.9 | 42.7 | 45.1 | 38.0 | 36.1 | 39.9 |

as the number of slow-frames) to represent target objects in different stages of videos, and evaluate our method on short and long videos. As shown in Tab. 6, compared with the decrease in accuracy on short videos, the accuracy on long videos has improved, which is worthy of further exploration.

## 6 CONCLUSION

In this work, we focus on the temporal reasoning during video segmentation. Through constructing a large-scale temporal-reasoning dataset and method, we fill the gap that how video reasoning methods handle queries with abundant temporal dynamics. The remarkable accuracy improvements fully demonstrate the necessity of temporal reasoning. The preliminary exploration of long-term video reasoning segmentation verifies the effectiveness of our method beyond keyframe selections. The introduction of temporal queries greatly expands the exploration space of the VRS field.

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

## A  APPENDIX

### A.1  THE USE OF LLMS

We use Large Language Models (LLMs) to aid or polish writing.

## A.2 STATISTICS AND ANNOTATION DETAILS

**Statistics of our proposed TempVRS dataset.** We compare the statistics of our proposed TempVRS dataset with existing referring/reasoning video segmentation dataset (open-sourced). Compared to these datasets, our TempVRS dataset has the most expressions reaching 200k, and the scale of videos has also reached the level of 30k. Meanwhile, our expressions rich in temporal dynamics have an average word length of 21.57, which exceeds the average length of these common expressions in existing video segmentation datasets. Besides, we additionally construct a long-term video zero-shot evaluation benchmark which contains 200 long-term videos and 2243 expressions, and the mean and max frame reach 1248 and 5574 respectively. Compared to short-term videos selected from SAV dataset (Ravi et al., 2024), these long-term videos cover a wider range of categories such as movies, documentaries, cooking, sports, etc. Meanwhile, long-term videos contain types of objects not seen in short-term videos such as cooking pan, test-tube used in appropriative scenarios. Therefore, whether from the perspective of video categories or object categories, these long-term videos are suitable to construct a zero-shot evaluation benchmark.

Table A1: Statistics comparison between our proposed TempVRS dataset and existing referring/reasoning video segmentation datasets. − denotes that we cannot make statistics on this item due to subsets of datasets are not open-sourced

| Dataset | Video | Object | Expression | Mean Frame | Max Frame | Masks | Avg. Len. |
|---|---|---|---|---|---|---|---|
| A2D Sentence (Gavrilyuk et al., 2018) | 3782 | 4825 | 6656 | 3.2 | 84 | 58k | 6.98 |
| J-HMDB Sentence (Gavrilyuk et al., 2018) | 928 | 928 | 928 | 34 | 40 | 32k | 6.16 |
| DAVIS$_{17}$-RVOS (Khoreva et al., 2019) | 90 | 205 | 1544 | 69 | 104 | 14k | 6.43 |
| Refer-YouTube-VOS (Seo et al., 2020) | 3978 | 7451 | 15009 | 27 | 36 | 131k | 9.68 |
| MeViS∗ (Ding et al., 2023) | 2006 | 8171 | 28570 | 79 | - | 443k | 7.07 |
| Long-RVOS (Liang et al., 2025a) | 2193 | 6703 | 24689 | 362 | - | 2.1M | - |
| ReVOS (Yan et al., 2024) | 1042 | 9084 | 35074 | 112 | 931 | 273k | 10.48 |
| Ref-SAV (Yuan et al., 2025) | 37311 | 72509 | 72509 | - | - | 6.0m | 83.6 |
| TempVRS(short-term) | 31701 | 68150 | 200040 | 83 | 653 | 4.0m | 21.57 |
| TempVRS(long-term) | 200 | 692 | 2243 | 1248 | 5574 | 87k | 34.2 |

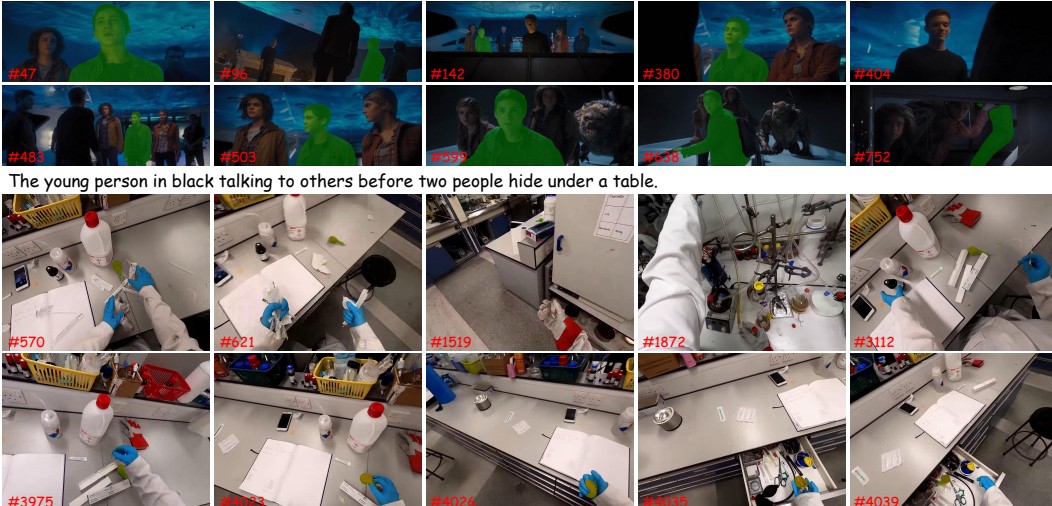

The young person in black talking to others before two people hide under a table.

The balloon on the table before a hand pick it up, squash it and put it into the drawer.

Figure A1: Examples of long-term videos including ego and exo perspectives for zero-shot evaluation. The number of these frames are marked in red.

**Annotation of long-term videos.** We select long-term videos from video object segmentation/tracking datasets including (Hu et al., 2023; Tang et al., 2023a; Hong et al., 2024; 2023). To avoid data contamination, we manually ensure these long-term videos are strictly orthogonal to existing referring/reasoning video segmentation datasets. For these videos selected from tracking datasets, we firstly unify the number of samples per second to 6 frames (e.g., from 30 to 6 fps),

then we generate mask annotations with SAM2 (Ravi et al., 2024) based on the bounding boxes in these tracking datasets and manually correct these auto-generated masks. As long-term videos are usually composed of multiple scenes as shown in Fig. A1, during captions of target objects in these long-term videos with our dataset construction pipeline, we firstly divide long-term videos into scene-clips with PySceneDetect (Castellano, 2025), which adds an additional temporal-scale (Frame/Clip/Scene/Video levels) in Step 2 to avoid losing details of target objects.

### A.3 TRAINING AND INFERENCE DETAILS

**Training and evaluation details.** During training, we first resize the input slow-frames to $448 \times 448$, then encode these frames into 256 tokens, fast-frames are encoded and compressed to 64 tokens. We decode the embedding of special token <TEP> with three-layers MLP. Following Sa2VA (Yuan et al., 2025), we adopt XTuner codebase for training and testing. The initial learning rate is set to $4e - 5$ and the number of uniform sampling slow-frame and fast-frame are set to 10 and 20. For fair comparison, we set the same context window of $K = 10$, frame size of $448 \times 448$, uniform-sampling and mask propagation strategies when conducting evaluations of Sa2VA (Yuan et al., 2025) and our method using the same backbone. More details can refer to our open-source code.

**Training strategies for distribution imbalance.** To address the distribution imbalance of existing referring/reasoning video segmentation datasets (Seo et al., 2020; Ding et al., 2023; Yan et al., 2024), we adopt several data augmentation strategies to balance the number of positive and negative samples during training with existing video segmentation datasets. Specifically, for the case that target objects have frames disappear/being obscured in a video (have negative samples), we adopt frame expansions (Cheng et al., 2021; Oh et al., 2019) to augment negative samples. For the case that target objects appears completely in every frame of a video (have no negative sample), we erase the target objects in the frame based on the ground truth mask of the target object, and inpaint background/paste other objects on the original object area to create negative samples. During these data augmentation process, we simultaneously adjust the spatiotemporal distribution of video frames to match frame-augmented videos.

**Inference details.** Following existing video reasoning segmentation methods, we extract the hidden state of the last layer in LLM to decode <SEG> by generating language prompts, then input these language prompts to generate masks with SAM2 decoder. During multi-node propagation, we select top-k clip based on the predicted correlation decoded from the embedding of special token <TEP>.

Table A2: Ablation studies on different top-k settings in multi-node propagation

| Setting | Val (Short) | | | Val (Long, zero-shot) | | |
|---|---|---|---|---|---|---|
| | $\mathcal{J}\&\mathcal{F}$ | $\mathcal{J}$ | $\mathcal{F}$ | $\mathcal{J}\&\mathcal{F}$ | $\mathcal{J}$ | $\mathcal{F}$ |
| $k = 3$ | 43.3 | 42.2 | 44.4 | 38.3 | 36.2 | 40.4 |
| $k = 5$ | 43.9 | 42.7 | 45.1 | 38.0 | 36.1 | 39.9 |
| $k = 7$ | 44.1 | 42.8 | 45.4 | 36.9 | 35.3 | 38.5 |

Based on the ablation studies as shown in Tab. A2, we set $k = 5$ to achieve balanced performance on both short-term and long-term evaluation sets. When facing long-term videos and adopting the strategy of generating multiple <SEG>, we share the same <SEG k> for the frames in the same $k - th$ clip to act as prompts for mask decoder. During mask propagation, we keep the same direction of propagation from multiple nodes to make full use of SAM2 memory.

