# OpenReview forum: "More Than a Snapshot: Forcing Temporal Reasoning in Video Segmentation"
_ICLR.cc/2026/Conference — ICLR 2026 Conference Withdrawn Submission_

### Official Review · Reviewer_VDM6 · 2025-10-19

**Soundness:** 3
**Presentation:** 2
**Contribution:** 3
**Rating:** 4
**Confidence:** 4

**Summary:**

The authors constructed a large-scale temporal video reasoning segmentation dataset containing 30,000 videos and 200,000 queries, where temporal reasoning based on video understanding is critical for identifying target objects. To solve the specific problem The authors proposed a temporal reasoning video segmentation method leverages spatio-temporal reasoning capabilities of MLLMs by learning spatiotemporal distributions. A fast-slow frame sampling strategy is introduced to uniformly sample frames and mitigate potential biases in frame selection. The method achieves state-of-the-art performance on multiple datasets while utilizing a lighter-weight MLLM under the same evaluation protocol.

**Strengths:**

• The paper’s motivation is clearly articulated and logically justified through well-designed experimental analysis.
• A large-scale dataset is constructed to address the target problem, providing a foundational resource for advancing community research in this domain.
• The proposed solution is simple yet effective, achieving impressive results with smaller MLLMs.

**Weaknesses:**

• The manuscript’s descriptions are insufficiently clear. Despite the simplicity of the proposed pipeline, the technical specifics of its components remain inadequately explained.
• The experiments are not conducted and explained thorough enough, see questions.

**Questions:**

• Table 3 Presentation: Why are the experimental results on the TempVRS dataset in Table 3 still split into two rows? What does "additionally trained with our TempVRS dataset" imply when benchmarking on the same dataset?
• Technical Implementation: The paper proposes a "clip-wise supervision" strategy. Could the authors clarify how this is technically implemented? Furthermore, does the choice of clip size significantly impact the performance?
• Ablation Study Limitations: The ablation study for the  token lacks a baseline comparison (e.g., removing both the  token and its supervision). Since the  token does not explicitly participate in the segmentation process, what is the underlying mechanism by which it enhances segmentation performance?
• Generalization Concerns: The ablation experiments are conducted exclusively on the proposed TempVRS dataset, where the results align well with the paper’s motivation. However, TempVRS heavily emphasizes temporal information. Would the same trends persist on other datasets (e.g., those with less temporal dependency)? If not, does this suggest limited generalization capability of the proposed method in broader, open-ended scenarios?

---

### Official Review · Reviewer_dc4i · 2025-10-26

**Soundness:** 3
**Presentation:** 3
**Contribution:** 2
**Rating:** 4
**Confidence:** 4

**Summary:**

This paper identifies a gap in Video Reasoning Segmentation (VRS), noting that existing benchmarks lack queries requiring temporal reasoning (understanding changes over time) and current methods often fail by just analyzing isolated keyframes. It introduces a new, large-scale dataset (30k videos, 200k queries) specifically designed to include tasks that demand temporal reasoning. Additionally, It proposes a new temporal video reasoning method. This approach avoids the flawed keyframe selection process by interleaving uniformly sampled frames and explicitly injecting spatiotemporal information.

**Strengths:**

1. This paper constructs a large-scale Temporal Video Reasoning Segmentation dataset containing 30k videos and 200k queries.
2. This paper proposes a benchmark to address a critical gap in video temporal segmentation, particularly for evaluation of temporal reasoning.

**Weaknesses:**

1. Lack of Dataset Construction Details: The paper omits crucial details about the dataset's construction. For reproducibility and a clear understanding of the data's scope, please provide specifics on the prompts used for query generation and the exact filtering mechanisms employed by human annotators.

2. Limited Query Diversity: The template-based generation approach is a concern, as it likely limits the linguistic and structural diversity of the queries, which could bias the benchmark.

3. Insufficient Scale for Evaluation: The dataset's scale, particularly the 1000 videos (and only 150 long videos) for assessment, seems insufficient for robustly evaluating and comparing model performance, especially for complex reasoning.

**Questions:**

See Weaknesses.

---

### Official Review · Reviewer_3txc · 2025-10-31

**Soundness:** 3
**Presentation:** 2
**Contribution:** 2
**Rating:** 2
**Confidence:** 4

**Summary:**

This paper aims to improve the temporal reasoning capability in video segmentation. It first constructs a temporal-reasoning dataset emphasizing temporal order, event, and count. Then, it proposes a MLLM-based video reasoning segmentation model with a slow-fast frame interleaved design. The experiment shows that the proposed method can achieve competitive performance on the proposed TempVRS dataset and existing benchmarks.

**Strengths:**

1. The paper proposes a dedicated dataset for temporal reasoning.
2. Subsequently, it designs a method to encourage temporal reasoning with a slow-fast interleaved arrangement.
3. The method achieves decent performance on the evaluated benchmarks.

**Weaknesses:**

1. There are already a lot of existing VOS datasets focusing on temporal dynamics, e.g., MeViS [1]. This paper proposes a new dataset, but does not clearly demonstrate the difference with existing works, making the motivation somehow weak.
2. The dataset construction pipeline is too empirical and lacks quality control. For example, are the temporal order, event, count defined by the authors or from some existing literature or empirical observations? What is the supporting evidence? Besides, the LLM-generated queries may not align well with the visual content due to hallucination. However, it seems that there is no quality control in this process.
3. Lack of novelty in method. The core design of the method is the slow-fast interleaved arrangement. However, this has been largely exploited in existing video reasoning segmentation works [2]. This paper does not discuss the relation and difference with existing methods.
4. Another design, SpatialTemporalDistribution, is not well presented in the current paper. It is not clear what does the SpatialTemporalDistribution means and how does the supervision undergo.


[1] MeViS: A Large-scale Benchmark for Video Segmentation with Motion Expressions.

[2] One Token to Seg Them All: Language Instructed Reasoning Segmentation in Videos.

**Questions:**

See weakness.

---

### Official Review · Reviewer_KjGA · 2025-11-01

**Soundness:** 3
**Presentation:** 2
**Contribution:** 3
**Rating:** 4
**Confidence:** 3

**Summary:**

This paper investigates the video reasoning segmentation task, putting address on the temporal reasoning and dynamics of videos. To facilitate this, this paper proposes a tempral reasoning segmentation dataset, named TempVRS, including 30k videos and 200k queries. This paper also proposes an optimized pipeline, including dense slow-fast frame sampling, clip-wise supervision, to overcome diffuculties of temporal reasoning segmentation task. Experiments show the proposed dataset TempVRS is challenging but the proposed method can outperform previous methods on this challenging task.

**Strengths:**

1. This paper investigate the video segmentation task, especially in the settings of strong temporal context, which is relatively underexplored before.

2. The paper contributes the dataset, TempVRS, as a training dataset and also a benchmark. More importantly, this paper details the data pipeline to collect and annotate queries of temporal context with LLMs.

3. This paper also address the shortcoming of keyframe selection-based methods and propose a slow-fast like method to address the memory and accuracy trade-off in the long video segmentation task.

**Weaknesses:**

1. Though being sound in experimental results, the presentation of this paper should be improved. It seems taking a lot of efforts to understand several sentences due to misleading wording & structures, e.g., line 95-96, `higher` in `makes higher demands for` should be `stricter`, line 283-285.

2. Related to weakness 1., the introduction and usage of <TEP> is super confusing. As a paper reader I cannot figure out clearly what it does and what it is designed for after reading the relevant texts twice. Also, the <TEP> is not depicted directly in Figure 4 & 5, which seems misleading. Also, what is spatio temporal distribution in Figure 4(b)? Is it the clip-wise supervision in Figure 5? The author is encouraged to put more efforts to describe these details with clear sentences and careful phrasing.

3. As a paper with a benchmark dataset and a method tailored for the proposed benchmark, the author is highly encouraged to include more details of results in Table 3. For example, on which of temporal order, event and count, the proposed method perform the best and worst, respectively? A more granular comparison with previous methods would also be valuable. In addition, the experimental section should devote a larger portion to TempVRS results and their analysis, as TempVRS is the paper’s contribution after all.

**Questions:**

See Weaknesses.

---

### Note · Authors · 2025-11-24

I have read and agree with the venue's withdrawal policy on behalf of myself and my co-authors.